# Stress, Inflammation and Metabolic Biomarkers Are Associated with Body Composition Measures in Lean, Overweight, and Obese Children and Adolescents [note 1]

**DOI:** 10.3390/children9020291

**Published:** 2022-02-21

**Authors:** Eirini V. Christaki, Panagiota Pervanidou, Ioannis Papassotiriou, Despoina Bastaki, Eleni Valavani, Aimilia Mantzou, Giorgos Giannakakis, Dario Boschiero, George P. Chrousos

**Affiliations:** 1Childhood Obesity Clinic, First Department of Pediatrics, School of Medicine, National and Kapodistrian University of Athens, “Aghia Sophia” Children’s Hospital, 11527 Athens, Greece; ppervanid@med.uoa.gr (P.P.); amantzou@med.uoa.gr (A.M.); chrousge@med.uoa.gr (G.P.C.); 2Unit of Developmental and Behavioral Pediatrics, First Department of Pediatrics, School of Medicine, National and Kapodistrian University of Athens, “Aghia Sophia” Children’s Hospital, 11527 Athens, Greece; bastakidespoina@gmail.com (D.B.); elvalav@yahoo.com (E.V.); 3Department of Clinical Biochemistry, “Aghia Sophia” Children’s Hospital, 11527 Athens, Greece; ipapassotiriou@gmail.com; 4Computational Biomedicine Laboratory, Institute of Computer Science, Foundation for Research and Technology Hellas (FORTH), 70013 Heraklion, Greece; ggian@ics.forth.gr; 5Institute of AgriFood and Life Sciences, University Research Centre, Hellenic Mediterranean University, 71410 Heraklion, Greece; 6BioTekna Co., 30020 Venice, Italy; dario.boschiero@gmail.com; 7University Research Institute of Maternal and Child Health and Precision Medicine and UNESCO Chair on Adolescent Health Care, 11527 Athens, Greece

**Keywords:** stress, childhood obesity, inflammation, body composition, bioimpedance, heart rate variability

## Abstract

The aim of this study was to examine the associations between multiple indices of stress, inflammation and metabolism vs. body composition parameters in 121 (43 boys, 78 girls) children and adolescents, aged 5–15 y. Subjects were divided into two groups: normal weight (N) (N = 40, BMI z-score = −0.1923 ± 0.6), and overweight/obese (OB) (N = 81, BMI z-score = 2.1947 ± 1.4). All subjects completed the State-Trait Anxiety Inventory for Children (STAIC) and Children’s Depression Inventory, and underwent cortisol measurements in hair, diurnal series of saliva, and morning serum. Circulating concentrations of high sensitivity C-reactive protein (hsCRP) and other inflammation biomarkers were also obtained. Body composition analysis was performed with a clinically validated, advanced bioimpedance apparatus (BIA), while heart rate variability (HRV) was measured as a stress biomarker by photoplethysmography (PPG). The OB group had a higher STAIC-state score, waist-to-hip ratio, skeletal muscle mass, and total and abdominal fat mass, and a lower percent fat-free mass (FFM) and bone density than the N group. HRV did not differ between the groups. In the entire population, percent fat mass correlated strongly with circulating hsCRP (r = 0.397, *p* = 0.001), ferritin, and other inflammatory biomarkers, as well as with indices of insulin resistance. A strong correlation between serum hsCRP and hair cortisol was also observed (r = 0.777, *p* < 0.001), suggesting interrelation of chronic stress and inflammation. Thus, body fat accumulation in children and adolescents was associated with an elevation in clinical and laboratory biomarkers of stress, inflammation, and insulin resistance. BIA-ACC and PPG can be utilized as a direct screening tool for assessing overweight- and obesity -related health risks in children and adolescents.

## 1. Introduction

Despite the growing literature on a possible relation between stress and obesity, most of the methods and interventions to prevent, assess, and treat obesity do not include efficient monitoring and treatment tools to estimate and eliminate the impact of stress on the progress of excessive weight gain. Diet, exercise, and medications are the most investigated and used treatments, but none of them have been totally effective as a sole treatment [1]. The multifactorial pathogenesis of childhood obesity is strongly associated with prolonged and excessive activation of the stress system [2,3,4]. 

There are two major components of the stress response: the autonomic nervous system (ANS) and the hypothalamic–pituitary–adrenal (HPA) axis. The main stress hormone associated with excessive weight gain and involved in metabolic dysregulation is cortisol, which may be hyper-secreted by the adrenal cortex as a result of chronic stress system activation [5]. Many studies have demonstrated correlations between long-term cortisol secretion levels [6], as measured in scalp hair, in both adults [7] and children [8], as well as in diurnal saliva samples and obesity [9]. 

Heart rate variability (HRV) is another increasingly used stress biomarker, as it is recognized as a quantitative marker of ANS activity [6,10]. One of the aims of this study was to further explore the interrelations between stress and HRV in this young population at risk for obesity-related morbidity. 

Cardiovascular outcomes and metabolic dysregulation associated with excessive body weight are not only correlated with stress indices, but also with low-grade systemic inflammation [11]. Serum levels of high-sensitivity C-reactive protein (hsCRP) are a direct inflammatory biomarker of childhood obesity-related inflammation [12], alongside other circulating markers, including ferritin, white blood cell count (WBC), and red blood cell width distribution (RDW) [13]. RDW predicts all-cause and cardiovascular (CVD) mortality and is strongly associated with inflammation deriving from excess body fat accumulation [14], while elevated WBC counts in obese children correlate with metabolic syndrome components [15]. Pro-inflammatory cytokines are also secreted by adipose tissue and are elevated proportionally to the degree of adiposity [16]. Excessive secretion of pro-inflammatory cytokines stimulates the HPA axis and increases cortisol production, which then stimulates an anti-inflammatory effect [17,18]. Consequently, there is crosstalk between the HPA axis and the inflammatory reaction and this may relate to the role of HPA axis alterations in the development of obesity and its complications [19,20].

Low-grade inflammation can be depressogenic and contribute to the disruption of glucocorticoid receptor function and expression, leading to unrestrained inflammatory responses that could further fuel depressive symptoms [21]. As stress and low-grade inflammation, as well as depressive symptoms [22], co-exist in childhood overweight and obesity, it is important to further explore the associations between them and more efficacious methods of evaluation and prevention of childhood obesity. The aim of this cross-sectional study was to examine the associations between primary mediators of chronic stress and secondary outcomes of the stress response, particularly inflammatory and metabolic biomarkers and body composition in children and adolescents. One of the research aims was to investigate the interrelations between stress, inflammation and metabolic biomarkers, and direct adiposity in children and adolescents, as well as the correlations between these biomarkers and behavioral measures (anxiety and depression symptoms). Finally, it was examined whether Bioimpedance (BIA-ACC) and photoplethysmography (PPG) could add useful research and clinical information in the assessment of child and adolescent obesity.

## 2. Materials and Methods

### 2.1. Participants

In total, 121 children and adolescents (78 girls and 43 boys), aged 8.93 ± 2.23 years, were recruited from the Childhood Obesity Clinic of the First Department of Pediatrics, School of Medicine, National and Kapodistrian University of Athens, “Aghia Sophia” Children’s Hospital, Athens, Greece. This was a cross-sectional analysis of baseline parameters of children and adolescents who participated in a longitudinal life-style interventional study. Baseline evaluation included the assessment of psychosocial and anthropometric parameters, medical history, blood testing and complete data using Bioelectrical Impedance (BIA) and photoplethysmography (PPG). With the exception of one child, all participants were Caucasians. The participants of the study were divided into two groups: group N (N = 40) included normal weight participants and group OB (N = 81) included overweight and obese participants. Participants of the group N were recruited from the community by research advertisement, or from family members from the group of overweight and obese subjects. Exclusion criteria encompassed: chronic use of any medication, mental disorders or pre-existing psychopathology, chromosomal disorders affecting growth and puberty, underlying chronic illnesses, such as cardiovascular, rheumatologic and renal diseases. Ethical approval of the study was obtained from the Research Ethics Committee of “Aghia Sophia” Children’s Hospital (protocol number: 4737/date: 24 February 2014), and all procedures were in accordance with the standards of the responsible committee on human experimentation, and with the Helsinki Declaration [23].

### 2.2. Clinical Evaluation

The children underwent a standard clinical examination by a pediatrician, including assessment of their pubertal status. Waist circumference was measured with an inelastic tape, using the narrowest part of the trunk between the ribs and the iliac crest and the hip at the greater trochanter level, whereas body weight was measured using the Inbody 320 (Biospace Co., Soul, Korea) with minimum clothing, and barefoot. Height was also measured barefoot, using a stable stadiometer (SECA 213). BMI was calculated as the children’s’ body weight in kilograms divided by the square of their height in meters. BMI z-scores were calculated based on the Greek Growth Charts [24].

Anthropometric parameters [25] (height, weight, waist and hip circumferences) were collected after a 12-h fasting (overnight) between 8:30 and 10:00 a.m. Body composition data were obtained by Multifrequency Bioimpedance Analysis (Biotekna srl. Venice, Italy). The specific full-body hand-to-foot multifrequency bioimpedance (BIA) device that has been used in previous studies [26] uses two different frequencies, 1.5 kHz and 50 kHz, to estimate the levels of Total Body Water and Extracellular Water. Through the estimation of Fat Mass Percentage (FMP) and Extracellular Water, the device can calculate the Fat Mass (FM) and Fat Free Mass (FFM), both in kilograms and as a percentage, Skeletal Muscle Mass, both in kilograms and as a percentage, as well as Intracellular Water. The BIA device can also estimate Abdominal Adipose Tissue, Body Density—and Visceral Organ Mass in Kg. 

### 2.3. Hair Sampling

Hair samples were obtained with scissors from the posterior vertex as close to the scalp as possible. 1 cm of hair was obtained from each subject and stored in paper folders at room temperature until analysis. Considering that hair grows 1 cm per month, this quantity was obtained to evaluate hair cortisol levels representative of the last 1 month [27]. Samples weighted approximately 20 mg and were placed in grinding tubes (Precellys Lysing Kits, Bertin Technologies) followed by their lysis at 5000 rpm for seven cycles of 1 min each using the homogenizer by Minilys, Bertin Technologies and Precellys lysing kit tubes (tissue grinding CKMix50-R). Then, 1 mL of methanol 99.8% (Euroclone, Italy) was added into the tubes and the powder-form hair was extracted at room temperature with shaking for 16 h. The tubes were centrifuged using the Biofuge 13 (Heraeus Instruments) at 10,000 g for 5 min, the extract (700 μL) was transferred to a glass tube and the methanol was left at room temperature for evaporation until the samples were completely dried (72 h). Samples were then reconstituted in 100 uL phosphate-buffered saline (pH 8.0, 1 × PBS) and were vortexed for 1.5 min. Before analysis, samples were vortexed again. Finally, samples were analyzed by using automated Electrochemiluminescence immunoassay (ECLIA) “Cortisol II” on the automated analyzer Cobas e411-ROCHE DIAGNOSTICS (GmbH, Mannheim). The limit of detection, as reported by the manufacturer’s directions, was 0.054 μg/dL. 

### 2.4. Saliva Sampling

Participants collected their salivary samples in salivettes with the help of their parents. The parents were given written instructions on the methodology of collecting the salivary samples at home, on the closest Sunday to the visit at the clinic. Six salivary samples were collected throughout one day with the first sample collected at about 8.00 and 30 min later at about 8.30, and at 12.00, 15.00, 18.00 and 21.00. Every sample was collected after a minimum 30 min abstention from any food or drink intake. Participants were instructed to chew the salivette’s synthetic swab for 1 min, store it in a plastic tube (Sarstedt, Germany) and keep it at 0–4 °C. Parents were instructed to bring the samples in coolers at the clinic. The samples were then centrifuged at 3000 rpm for 5 min and stored at −80 °C, until the day of analysis. Cortisol was measured on the automated analyzer Cobas e411-ROCHE DIAGNOSTICS (GmbH, Mannheim) by electrochemiluminescence immunoassay (ECLIA). Diurnal salivary cortisol slope was calculated by subtracting lower from higher value (peak value of the day) and dividing by the number of hours separating the two samples, as implemented in previous research [28]. Cortisol total output was summarized using the area under the curve with respect to the ground (AUCg), calculated using the concentrations of the six serial salivary samples.

### 2.5. Blood Sampling

Blood samples were collected after a 12 h fasting (overnight) between 8:30 and 10:00 am, as applied in previous research [29]. Blood chemistry included measurements of serum fasting glucose, insulin, cortisol, total cholesterol (TC), high-density lipoprotein cholesterol (HDL), low-density lipoprotein cholesterol (LDL), triglycerides, ferritin, alanine aminotransferase (SGPT), aspartate aminotransferase (SGOT) and gamma-glutamyltransferase (γGT). Measurements were performed using the Cobas 6000 Clinical Chemistry analyzer (Roche diagnostics). Serum cortisol and insulin measurements were performed based on electrochemiluminescence immunoassay principle of Cobas e 411 analyzer (Roche Diagnostics). Ferritin levels were determined by means of an electrochemiluminescence technique, using the Roche E411 Cobas immunoassay analyzer (Roche Diagnostics), as used in previous research [30].

For the assessment of the obesity-induced low-grade inflammation, we used the high sensitivity C-reactive protein (hsCRP) and ferritin as biomarkers. Serum hsCRP concentrations were measured using the BN ProSpec nephelometer (Siemens Healthineers, Erlangen, Germany) with fully automated latex particle-enhanced immunonephelometric assays. The intra-assay and inter-assay CV’s were <6% and <7%, respectively. Serum hsCRP > 10.0 mg/lt were excluded from the analyses. 

### 2.6. Heart Rate Variability Measurements

Resting heart rate was obtained using a multi-channel measurement photoplethysmograph technology (PPG, Biotekna-Venice, Italy) applied to the distal ends of the limbs. Sensors were placed on both index fingers of the subjects, who remained seated for 20 min before the measurement. Data collection from the PPG sensor lasted 5 min, while the subject was asked to remain seated in a quiet environment. It has been proposed that 5 min HRV parameters—especially low vagal parasympathetic activity—could serve as chronic stress indicators [31]. All measurements took place between 8 and 10 a.m. in a fasted state. Alongside resting heart rate, the PPG device measured the standard deviation of NN intervals (SDNN), High Frequency activity (HF), Low Frequency activity (LF) and scatter area. SDNN is the standard deviation of the average beat-to-beat intervals and is a measure of HRV. The sympathetic nervous system (SNS) can increase the conductivity of the cell membranes of the heart cells that control the heart rate and lead to a higher heart rate, while the parasympathetic nervous system has the opposite effect, leading to a lower heart rate. HRV provides an insight into both autonomic nervous system limbs and their interactions, the so-called sympatho-vagal balance. It is generally accepted that the activities of the SNS and PNS are reflected in the LF and HF band, respectively [32], while their ratio—LF/HF—has been used to quantify the degree of sympatho-vagal balance. A new metric has been proposed recently, the 2D scatter plot of LF vs. HF, which utilizes all available information in LF and HF power and can even contribute to the categorization of the stressor [33].

### 2.7. Questionnaires

All children completed the Children’s Depression Inventory and the State-Trait Anxiety Inventory for Children (STAIC), developed by Spielberger et al. in 1973 [34] and tested for its validity and reliability in the Greek population by Psychountaki et al. in 2003 [35].The STAIC inventory consists of 40 self-report questions with a 3-point scale: 20 of them focus on the child’s current level of anxiety (state), and the rest focus on the child’s general anxiety level (trait). This self-report questionnaire is used to distinguish between a general tendency to anxious behavior as an internal characteristic of the personality, and anxiety, as a fleeting emotional state. Children’s Depression Inventory (CDI) is a self-report test that helps to diagnose cognitive, affective, and behavioral signs of depression in children and adolescents. CDI has been tested for its validity and reliability in the Greek population by Giannakopoulos et al. [36]. 

Both questionnaires are self-report instruments that are subject to self-report biases. 

### 2.8. Statistical Analyses 

Statistical analyses were carried out using the Statistical Package for the Social Sciences (SPSS, version 21; SPSS Inc., Chicago, IL, USA) and Matrix Laboratory (MATLAB v.2018b); significance level α was set at 0.05. Descriptive statistics were presented as mean ± SD for continuous variables or as percentages for categorical variables (Table 1). 

The variables under investigation were initially checked for normal distribution using the Kolmogorov–Smirnov test. Statistically significant differences between groups (normal weight, overweight/obese) were evaluated using independent samples Student *t*-test controlling the means difference. The problem of multiple comparisons was adjusted using the False Discovery Rate (FDR) method, which is considered a consistent adjustment method [37]. For the evaluation of cortisol concentration during the different measures of the day, a Repeated Measures ANOVA was performed; post hoc analysis (Bonferroni criterion) was also checked in order to reveal the pair differences along the hour of the day and the interaction between diagnostic groups. 

Correlations between parameters under investigation were evaluated by Pearson’s cross-correlation or by the nonparametric Spearman test, depending on investigated variables, adjusted for sex and Tanner stage. Categorical data were evaluated by Pearson’s chi-square test. For all the tests, the significance level α was set to 0.05. The prevalence of excess body fat, including overweight and obesity, was defined using the BMI-SDS cut-off points of the International Obesity Task Force (IOTF) criteria (88th percentile in girls and 90th percentile in boys) [38]. In order to calculate the BMI z-scores the recent Greek growth charts [24] were used.

## 3. Results

Participants’ sociodemographic (age, sex), anthropometric (BMI z-score, Tanner pubertal stage), psychometric, body composition and bioimpedance variables were estimated and checked for their normal distribution. The overweight and obese participants were grouped together as they presented similar profiles in body composition and bioimpedance variables. Statistical differences between groups (normal weight, overweight/obese) were assessed using independent samples *t*-test and the results are summarized in Table 1. 

There were no significant differences between groups in age, Tanner stage, percentage of boys and girls, levels of exercise and screen time. Socioeconomic status assessed through parents’ education and family income was similar in the two groups. The dataset was selected in such a way so as not to cause significant differences between the two groups on the above parameters, as well as to minimize a bias effect on stress, inflammation, body composition and metabolic biomarkers, or other confounding factors.

As expected, BMI z-score, waist-to-hip (WtH) ratio and body composition analysis values were significantly different between normal weight and overweight/obese subjects (*p* < 0.0001). Unsurprisingly, a strong positive correlation between BMI z-score with abdominal adipose tissue (r = 0.938, *p* < 0.0001) was observed as well.

HCC, serum cortisol, AUCg, insulin, WBC, RBC, RDW, HCt, Iron, Ferittin, glucose, hsCRP, FMP were initially evaluated for group differences adjusting multiple comparisons with False Discovery Rate (FDR) [37], which is considered a well-established adjustment method [37]. The alpha-level according to the FDR was calculated as
*a_i_* = (*i*/*c*) · *α_FW_*
where *α_FW_* is the Familywise error rate, *c* is the total number of comparisons, and *i* is the comparison currently being made on the ascending sorted *a_i_*. The results are presented in Table 2.

As depicted in Table 2, there were statistically significant differences between the groups for the variables FMP (F = 88.35, *p* < 0.001), and WBC (F = 7.76, *p* = 0.008).

The overall salivary cortisol concentration had its maximum value 12.2 ± 0.7 (mean ± SD.) at 8:00 a.m. and decreased to 4.4 ± 0.6 at 21:00 p.m.

We checked the salivary cortisol concentration variation in the two groups (normal weight and overweight/obesity) and the levels are presented in Figure 1.

The different serial diurnal samples elicited statistically significant changes in salivary cortisol concentration over time (repeated measures ANOVA, F(2.691, 261.02) = 47.63, *p* < *0*.001). Post hoc analysis revealed that the morning hour (08:00 and 08:30) samples were higher than the measures later in the day. However, there was no interaction between diagnostic groups and timing of the salivary cortisol. 

### Correlation Analyses

Correlation analyses were performed to reveal the interaction of inflammation, stress (HCC, AUCg, HRV measures), and metabolic dysregulation biomarkers under investigation, either among themselves or with respect to the two groups (normal weight, overweight/obese) according to the study’s second and third research hypotheses. 

Resting heart rate was positively correlated with morning salivary cortisol (r = 0.217, *p* = 0.024), but there was no correlation of resting heart rate with BMI z-score (r = 0.061, *p* = 0.508) or with FMP (r = −0.090, *p* = 0.332) or with any other variables among those investigated. Salivary cortisol and RHR were correlated with each other, but not with excessive body fat accumulation. SDNN (r = 0.279, *p* = 0.008)—a measure of HRV and scatter area (r = 0.294, *p* = 0.005)—which reflects ANS activity, correlated with STAIC-state scoring. Moreover, there was a negative correlation of waist-to-hip-ratio with scatter area (r = −0.237, *p* = 0.032) and of waist circumference with SDNN (r = −0.229, *p* = 0.039). No other correlations between PPG measurements and body composition, cortisol concentrations or psychometric variables studied were observed. STAIC-state scoring was positively correlated with FMP (r = 0.255, *p* = 0.014), as well as with salivary cortisol slope (r = 0.323, *p* = 0.002), whereas there was no correlation between STAIC-trait and FMP or salivary cortisol slope.

As expected, in the entire study population, BMI z-score was positively correlated with fat mass percentage (FMP) (r = 0.843, *p* < 0.0001), as well as with WtH ratio (r = 0.533, *p* < 0.0001). There was a significant correlation of FMP with high sensitivity C-reactive protein (hsCRP) levels (r = 0.397, *p* = 0.001), as well as with ferritin (r = 0.305, *p* = 0.002) and insulin levels (r = 0.578, *p* < 0.0001). Moreover, red cell distribution width (RDW) (r = 0.186, *p* = 0.034), WBC (r = 0.386, *p* = 0.007), salivary cortisol slope (r = 0.223, *p* = 0.03), uric acid levels (r = 0.431 *p* < 0.0001), triglycerides (r = 0.366, *p* < 0.0001), SGPT (r = 0.235, *p* = 0.021) and γGT levels(r = 0.387, *p* < 0.0001) correlated positively with FMP, when controlled for Tanner stage and sex, whereas iron levels (r = −0.298, *p* = 0.004) and HDL-C (r = −0.339, *p* < 0.0001) correlated negatively with FMP, confirming the hypotheses of the study that inflammation and stress biomarkers are associated with obesity related metabolic dysregulation. Differences between groups in cortisol concentrations and the above-mentioned blood chemistry indices are presented in Table 3.

FMP correlated positively with hair cortisol concentrations (HCC) (r = 0.253, *p* = 0.018), as well as with the AUCg of salivary cortisol levels (r = 0.283, *p* = 0.018); there was no correlation of serum cortisol with any of the above-mentioned markers. It is worth mentioning that there was no statistically significant difference of HCC between normal weight and overweight/obese participants. The different temporal samples elicited statistically significant changes in salivary cortisol concentration over time using repeated measures ANOVA, *F*(2.691, 261.02) = 47.63, *p* < 0.001. However, there was no interaction between diagnostic groups and timing of the salivary cortisol concentrations, as shown in Figure 1. 

Investigation of the interaction between inflammation and stress biomarkers revealed a strong correlation of HCC with hsCRP (r = 0.777, *p* < 0.001), which is shown in Figure 2. Subjects exhibiting low grade inflammation (as assessed by levels of hsCRP > 2.5 mg/L) present a strong correlation of HCC with WtH ratio (r = 0.793, *p* = 0.019) and with CDI scoring (r = 0.689, *p* = 0.040). 

Moreover, ECW percentage was negatively correlated with hsCRP (r = −0.321, *p* = 0.011), as well as with ferritin (r = −0.213, *p* = 0.037) and hair cortisol concentrations (r = −0.322, *p* = 0.002). White blood cells (WBC) were positively correlated with BMI z-score (r = 0.197, *p* = 0.032), with hsCRP (r = 0.257, *p* = 0.027), and with STAIC-state scoring (r = 0.195, *p* = 0.047).

## 4. Discussion

Although reports on body fat accumulation, mediators of the stress response, and systemic low-grade inflammation are scarce in children and adolescents, this relation has been widely reported in adults. In the present study, a strong correlation of FMP and inflammation markers, such as hsCRP, WBC, RDW, and ferritin, as well as metabolic dysregulation markers, such as fasting insulin, triglycerides, uric acid, SGPT, γGT, iron and HDL cholesterol, was observed in the entire population. Moreover, this study revealed a correlation of FMP with hair cortisol concentration and hsCRP levels. In addition, resting heart rate (RHR) was positively associated with morning salivary cortisol concentration in our subjects. It should be noted that instructions were given to the parents/caregivers to sample saliva at specific times (8:00, 8:30, 12:00, 15:00, 18:00, 21:00), which probably were not always aligned with the awakening (e.g., a child wakes up at 07:00 and has increased salivary cortisol levels at 07:30). 

RHR has been used to evaluate acute stress in obese subjects [39], with previous research showing elevated RHR in obese children and adolescents [40]. Basal heart rate has also been found elevated in depression, where high salivary cortisol concentrations indicate chronic stress and impaired HPA axis activity, with a compromised ability to maintain homeostasis [41]. HRV has been suggested as a stress marker in children as well [42]. In the present study, HRV was negatively associated with waist circumference, while the scatter area, an index of balanced ANS cardiac regulation by the ANS, was negatively associated with waist-to-hip ratio. Previous studies have demonstrated a positive association of HRV with adiposity. Most of these studies, however, used BMI as a measure of excess body weight, a parameter that does not differentiate between fat mass and fat free mass or body fat distribution [31]. On the other hand, visceral adiposity, but not BMI, has been associated with drop in HRV indices of cardiac parasympathetic activity in children [43]. Our findings are in accordance with these observations, while it has been suggested that low HRV (lower parasympathetic activity) might serve as a stress indicator in children [6,42], as well as an indicator of increased risk of future severe cardiovascular pathology [44]. 

STAIC-state scoring mostly reveals a general proneness to anxious behavior. In our study, STAIC-state was positively associated with FMP, SDNN, and scatter area, probably revealing a higher ANS activity in children with a tendency to anxiety. Although there was no significant difference in STAI scoring between groups, there was a correlation of STAIC-state with FMP, indicating an association of body fat accumulation and anxiety symptoms. In subjects with low grade inflammation, as assessed by levels of hsCRP > 2.5 mg/L, a strong correlation of HCC with Waist-to-Hip ratio and with CDI scoring was observed. CDI is one of the most widely used instruments for assessing the presence and severity of depressive symptoms in children and adolescents. It is worth noting that stress has been linked with depressive symptoms [45], while HCC has been widely used as a marker of chronic stress in children [46,47], as it reflects chronic activation of the hypothalamic-pituitary-adrenal axis. It is worth mentioning that there was a correlation of CDI scoring with HCC only in subjects with higher levels of inflammation, showing the potential role of inflammation in the crosstalk between the HPA axis regulation and the presence of depressive symptoms.

The fact that the relation of stress and central obesity defined by Waist-to-Hip ratio is stronger in subjects with obvious low-grade inflammation suggests a need to further explore the link between body composition and primary stress and inflammation biomarkers. In our study, hair cortisol—a chronic integrated stress biomarker—and hsCRP—a key inflammation biomarker—correlated positively with the FMP. Reciprocally, excess body weight is associated with systemic low-grade inflammation mediated by increased adipokine secretion, which can chronically stimulate the stress system [48]. The mechanisms triggering HPA axis activation and inflammatory responses in overweight and obesity are not completely understood as yet. It has been suggested that obesity is associated with an impairment of adipocyte metabolism, accompanied by recruitment of macrophages into the adipose tissue, causing local and systemic inflammation, and inducing production of acute-phase proteins, such as CRP [11]. 

In addition, the slope of salivary cortisol was positively correlated with FMP, a finding that contradicts previous findings in studies in adults that have linked a flat cortisol slope with stress-related dysregulation of circadian mechanisms and negative health outcomes [49]. The relation of psychosocial stress and diurnal cortisol regulation has not been extensively studied in children and adolescents. Chronic dysregulation of the stress system may be attributed to either over- or under-secretion of cortisol. In this study, an elevated cortisol slope was associated with greater body fat accumulation, which may have contributed to higher levels of inflammatory biomarkers, such as hsCRP and ferritin. It has been suggested that stress that occurs during a sensitive developmental period, i.e., when immune function is highly malleable, can affect the chronic functioning of the cells that regulate inflammation [50]. Immune responses can alter neural and endocrine function, while both neural and endocrine functions can affect the immune system. The dysregulation of HPA axis that occurs when the stress load is high, has the potential to bring about a systemic inflammatory state. It is unclear, therefore, whether stress is the mediator or the cause of the relation between excessive body fat accumulation and inflammation in children and adolescents.

Total diurnal salivary cortisol secretion, as measured with the AUCg, was positively associated with hair cortisol concentration. Our results are in accordance with previous research, showing a positive association between hair cortisol and salivary cortisol concentrations [9], while there was no association between HCC or salivary cortisol concentration with serum cortisol. 

### 4.1. Stress, Non-Adaptive “Para-Inflammation” Biomarkers, and Body Composition

In the current study, the strong correlation between body composition indices and anthropometric parameters, such as BMI z-score and waist-to-hip-ratio, indicated that body fat assessment through our BIA device is a good screening and monitoring method for this young population. BIA-ACC measurements using the apparatus employed here had high reliability and a relatively strong agreement with DEXA measurements in evaluating body composition variables in a sample of adults in a recent study [51].

In the present study, BMI z-score was associated with FMP and Waist-to-Hip ratio, as expected. The association of HCC with FMP was not reflected in BMI z-score, also as expected, as there was no significant difference between the groups. Fasting insulin was also associated with bioimpedance analysis measures, such as FMP and muscle mass, while controlling for the possible confounding factors of sex and Tanner pubertal stage. The correlation of FMP with Waist-to-Hip ratio, waist circumference, uric acid, triglycerides, HDL-C, SGPT, and GGT were as expected and can be attributed to overweight/obesity, while the negative correlation with iron levels is consistent with previous research [52]. It remains unclear whether the low iron concentrations observed in obesity reflect an inflammatory state, and/or if obesity is a risk factor for true iron deficiency [53]. Besides, a negative association of hsCRP and ferritin with ECW was demonstrated. The negative correlation of ECW with inflammatory markers disagrees with previous findings in the adult population, where ECW was positively correlated with the levels of hsCRP, possibly due to salt and water retention caused by elevated cortisol levels [54].

These findings emphasized the need for an easy-to-use method to assess body composition and related health risks. Multifrequency bio-impedance analysis is an evolving method that can contribute to this assessment and has been shown to impact upon clinical practice and to contribute to decisions for lifestyle interventions [55]. Apart from the FMP, the fat-free mass and the fat-free mass percentage are body composition indices that can contribute to the assessment of the body composition of a child, as well as the risk of sarcopenia [56]. In this study, fat-free mass percentage (% of body weight) was significantly higher in the lean group than the overweight/obese group. In overweight/obese children, it is not known whether decreased muscle mass is the result of chronic stress-related hypercortisolism. Previous research has shown that obesity in young adults is associated with sarcopenia resulting in what has been called “sarcopenic obesity” [57]. Moreover, RDW was significantly higher in the overweight/obese than the lean group, while elevated RDW was associated with sarcopenia, an association that was particularly strong in adults who were overweight or obese [58]. 

### 4.2. Strengths and Limitations of the Study 

A major strength of this study was the simultaneous measurements of multiple behavioral and somatic indices of stress, metabolism, inflammation, and the percentages of body fat, muscle mass, bone, and other compartments of human body composition in children and adolescents. The findings revealed the need to shed light on the mechanisms that determine body composition in children, as chronic stress was previously shown to be such a mechanism in adults [59]. There is evidence suggesting that the upregulation of the HPA axis may influence eating behavior through pathways that prominently control dietary intake among obese individuals [60]. It has been proposed that psychological stress and anxiety and depressive symptoms in children may influence biological responses that can impact health outcomes, such as decreased skeletal muscle mass and central adiposity, through the mediation of chronically elevated cortisol [61]. The findings of this study confirmed that body fat accumulation in children and adolescents is associated with biomarkers of stress and inflammation and that psychosocial factors might be related to metabolic dysregulation in children [62]. Moreover, increased levels of body fat were associated with elevated prevalence of self-reported state anxiety, a finding that is in accordance with previous research linking state anxiety with obesity [63]. 

The current study combined the assessment of excessive body weight and changes in body composition, with stress, inflammation, and behavioral markers of anxiety and depression, in order to shed light on the possible underlying associations between obesity and psychosocial health [22] using multiple objective parameters, such as hair cortisol concentration and circadian salivary cortisol profiles in children and adolescents. 

One of the limitations of our study is the relatively small number of participants and the fact that overweight and obese subjects were examined as one group. They were included in one group because there were no significant differences between overweight and obese children as far as most of the biomarkers examined were concerned. Moreover, these analyses were cross-sectional and, thus, causality could not be imputed. Finally, the collection of salivary cortisol samples was not started strictly at awakening, due to poor compliance of the participants to diurnal sampling procedures. Thus, our conclusions regarding AUCg and slope should be interpreted with some caution as the first two salivary samples of the day were not obtained accurately. As a result, the cortisol awakening response [64] was not evaluated in the analyses.

The data of the present study confirm previous findings of pediatric studies associating overweight/obesity with nonadaptive, low-grade “para-inflammation” [65], cardiometabolic risk factors, and stress-related behavioral problems, and provide additional evidence suggesting that multifrequency bioimpedance analysis can be used for screening, evaluating, and monitoring health risks in children and adolescents. 

## 5. Conclusions

The fact that BMI z-score has limited capability to contribute to the assessment of important stress, metabolic and inflammatory risk factors and higher levels of anxiety and depression in children and adolescents, unravels the need to use more sensitive tools, such as BIA and, possibly, PPG. Higher levels of inflammation and stress biomarkers, as well as disordered mood symptoms, should be taken into account in health policy and lifestyle intervention programs focusing on tackling excessive weight in children and adolescents.

## Figures and Tables

**Figure 1 children-09-00291-f001:**
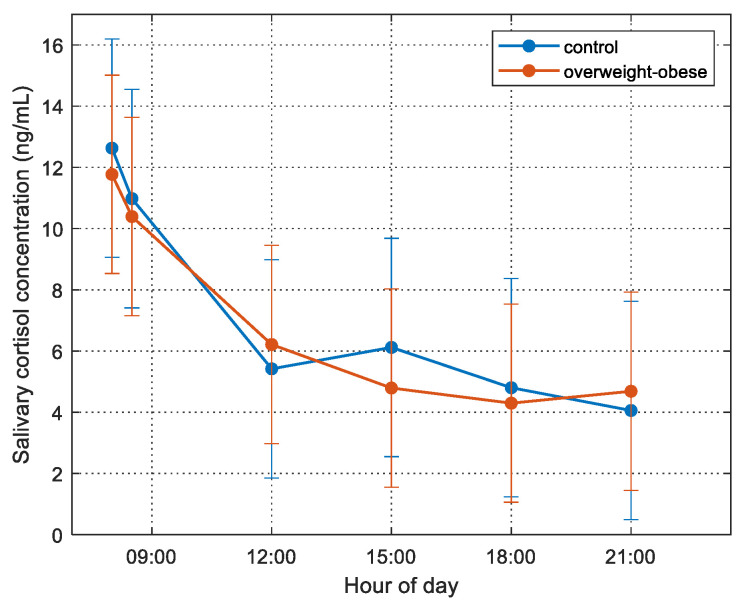
Cortisol concentration for the normal weight (blue line) and overweight/obese (red line) for the six different diurnal serial samples during the day of monitoring. Bars represent the std of the measurements.

**Figure 2 children-09-00291-f002:**
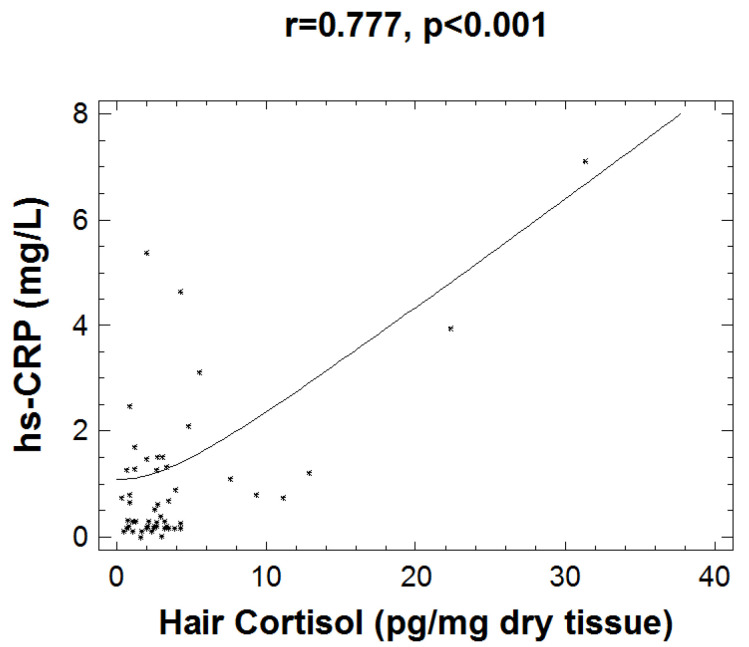
Graph depicting the positive correlation between hsCRP and hair cortisol measurements (r = 0.777, *p* < 0.001).

**Table 1 children-09-00291-t001:** Participants’ sociodemographic/anthropometric characteristics and statistical differences in body composition and bioimpedance variables between normal weight and overweight/obese group. Statistics are presented as means ± SD.

Parameter	Normal Weight (N = 40)	Overweight/ Obese (N = 81)	*p* Value
Age	8.74 ± 2.14	9.02 ± 2.28	0.522
Sex	70% Female 30% male	61.7% Female 38.3% male	0.375
BMI z-score	−0.19 ± 0.60	2.20 ± 1.43	**<0.001 ***
Tanner stage	90% pre-pubertal 6.5% mid pubertal 3.5% post pubertal	80.6% pre-pubertal 13.9% mid pubertal 5.6% post pubertal	0.279
Waist-to-Hip ratio (WtH)	0.85 ± 0.53	0.91 ± 0.61	**<0.001 ***
Levels of exercise (hours/per week)	6.16 ± 4.00	5.51 ± 3.29	0.359
Family income (EUR)	2.17 ± 0.66	1.89 ± 0.68	0.53
Parents’ education (years)	14.47 ± 2.30	14.29 ± 2.83	0.778
Screen time (hours/per week)	12.18 ± 9.6	15.6 ± 8.94	0.064
STAIC-state scoring	24.9 ± 4	27.96 ± 5.18	**0.002 ***
STAIC-trait scoring	30.23 ± 5.78	30.18 ± 6.21	0.969
CDI scoring	5.27 ± 4.66	6.01 ± 4.70	0.438
Total Body Water (% of body weight)	60.2 ± 8.38	48.47 ± 6.80	**<0.001 ***
Extracellular Water (ECW) (% of body weight)	52.58± 4.72	46.88 ± 5.05	**<0.001 ***
Intracellular Water (% of body weight)	47.43 ± 4.71	53.12 ± 5.05	**<0.001 ***
Fat Free Mass (FFM) (% of body weight)	89.9 ± 6.18	70.72 ± 7.48	**<0.001 ***
Fat Free Mass (Kg)	27.58 ± 6.32	34.91 ± 10.42	**<0.001 ***
Fat Mass (FM) (% of body weight)	10.1 ± 6.18	29.28 ± 7.48	**<0.001 ***
Fat Mass (Kg)	3.49 ± 3.17	15.07 ± 8.45	**<0.001 ***
Glycogen (% of body weight)	0.74 ± 0.13	0.81 ± 0.11	**<0.001 ***
Abdominal adipose tissue (% of body weight)	12.64 ± 7.75	36.59 ± 9.38	**<0.001 ***
Abdominal adipose tissue (Kg)	4.37 ± 3.97	18.84 ± 10.57	**<0.001 ***
Visceral organs tissue (Kg)	16.83 ± 4.47	15.89 ± 2.85	**0.046 ***
Skeletal muscle mass (Kg)	7.69 ± 2.79	11.03 ± 4.35	**<0.001 ***
Skeletal muscle mass (% of body weight)	27.59 ± 6.17	33.07 ± 5.66	**<0.001 ***
Body Density	1.06 ± 0.13	1.02 ± 0.15	**<0.001 ***
Phase angle	3.02 ± 0.49	2.92 ± 0.78	0.454
Resting heart rate (RHR) (pulses/min)	85.98 ± 11.5	83.97 ± 10.69	0.323
SDNN (ms)	133.026 ± 216.58	109.025 ± 121.78	0.441
Scatter area (ms²)	195.9 × 10^3^ ± 64.6 × 10^3^	80.5 × 10^3^ ± 383.1 × 10^3^	0.225
LF power	7.23 ± 1.63	7.44 ± 1.11	0.410
HF power	7.64 ± 1.88	7.76 ± 1.62	0.717
LF/HF ratio	0.833 ± 0.71	0.82 ± 0.66	0.920

Note: Bold type and * denote a statistically significant difference between groups (normal weight vs. overweight/obese).

**Table 2 children-09-00291-t002:** Comparison between normal weight and overweight/obese group regarding the variable’s cortisol (hair, serum), AUCg, insulin, blood parameters (blood white cells, blood red cells, red cell distribution width, HCt, iron, ferritin, glucose, etc.) adjusting with FDR.

Dependent Variable	*p*-Value	FDR Adjusted *a_i_*
AUCg	0.200	0.029
Hair cortisol concentration (HCC) (pg/mg)	0.917	0.046
Serum cortisol (mcg/dL)	0.697	0.033
hsCRP_(mg/L)	0.028	0.013
FMP (%)	**<0.001**	**0.004**
Insulin (μU/mL)	0.028	0.017
White blood cells count (WBC × 10^9^/L)	**0.008**	**0.009**
HCt (%)	0.719	0.038
Red cell distribution width (RDW %)	0.114	0.025
Iron (mcg/dL)	0.960	0.050
Ferritin (ng/mL)	0.093	0.021
Glucose (mg/dL)	0.823	0.042

**Table 3 children-09-00291-t003:** Blood chemistry, salivary and hair cortisol measurements in normal weight and overweight/obese participants. Mean values of each variable are presented along with standard deviations (Student’s *t*-tests).

	Normal Weight (N = 40)	Overweight/Obese (N = 81)	*p* Value
Hair cortisol concentration (pg/mg)	2.98 ± 5.41	3.16 ± 2.53	0.829
Morning salivary cortisol (first sample of the day) (ng/mL)	12.32 ± 5.79	11.79 ± 6.41	0.674
Red cell distribution width (RDW %)	13.45 ± 0.85	13.87 ± 1.11	**0.026 ***
White blood cells count (WBC × 10^9^/L)	6.72 ± 1.42	6.8 ± 1.72	0.811
Iron (mcg/dL)	98.1 ± 26.59	81.21 ± 24.85	**0.002 ***
Ferritin (ng/mL)	43.06 ± 22.44	49.40 ± 28.46	0.242
Serum cortisol (mcg/dL)	13.48 ± 6.07	12.39 ± 5.74	0.369
Insulin (μU/mL)	5.95 ± 2.75	11.87 ± 7.78	**<0.001 ***
Uric acid(mg/dL)	3.75 ± 0.62	4.37 ± 0.94	**<0.001 ***
Aspartate transaminase (SGOT) (U/L)	29.00 ± 13.11	27.01 ± 13.27	0.448
Serum glutamic pyruvic transaminase (SGPT) (U/L)	17.47 ± 9.17	25.19 ± 28.74	0.11
Gamma-glutamyl Transferace (γGT)(U/L)	11.51 ± 3.19	14.91 ± 6.05	**0.002 ***
Triglycerides (mg/dL)	52.84 ± 16.76	74.68 ± 44.94	**0.005 ***
Total Cholesterol (mg/dL)	166.32 ± 27.62	161.90 ± 29.6	0.441
Low-density lipoprotein (mg/dL)	93.73 ± 22.95	92.89 ± 25.3	0.863
High-density lipoprotein (mg/dL)	61.97 ± 11.28	54.58 ± 12.87	0.003

* and bold denote a statistically significant difference.

## Data Availability

The data presented in the current study are available in www.researchgate.net (accessed on 15 January 2022) since the 15 of January 2022. Doi:10.13140/RG.2.2.33953.33124.

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
