# Peer review of "Stress, Inflammation and Metabolic Biomarkers Are Associated with Body Composition Measures in Lean, Overweight, and Obese Children and Adolescentsâ€"

_children, 2022, doi:10.3390/children9020291_

Round 1
Reviewer 1 Report
It is a very interesting study about a relevant topic.
INTRODUCTION
The hypotheses should be written in a sole paragraph, not written in the first person.
METHODS:
-Add a reference for the anthropometric parameters and the tool used.
-Add a reference for blood sampling.
-Regarding the STAIC questionnaire, try to put all the information in a single paragraph.
-Depending on if the variables are parametric or not, you should use Pearson's correlation test or another test.
RESULTS:
-Please be more consistent with the number of decimals used in table 1. Include the abbreviations.
-Lines 274-277 in the same paragraph there is "as expected" twice.
-The title of Table 2 is not clear enough. Maybe it should be "Comparison between groups......". Include the abbreviations.
-Table 3: include abbreviations, remove de "P=" from the p-value column. Instead of "ns", put the significant results in bold letters or with an *. Il should be consistent between the 3 tables.
DISCUSSION
Instead of "in our study", use "in the present study, in this study...". Same comment on lines 439 and 445
REFERENCES
Some of the references are a bit old. Perhaps you could include some more current citations.
Reviewer 2 Report
This is a very interesting paper on relation of obesity, chronic stress and low inflammation in children and adolescents. Although it is a cross-sectional study therefore it cannot talk about causality, the paper is nicely and clearly written, methodology clear and precise. Moreover, results are presented in appropriate and easy-to-follow manner.
I only have few suggestions for the authors:
Section "Introduction"
- line 79 to 81 - the sentence is not clear
Section "Results"
- Table 1 - specify in the table or in the table heading that variables are presented as means with standard deviation
Section "Literature"
- a big proportion of literature is outdated. I would suggest for the authors to pick or leave only the studies with newer dates if that is possible
Reviewer 3 Report
The Authors have shown that in children, stress, inflammation and metabolic markers and adiposity are inter related. The manuscript is well written and the study is well designed. The tables are clear to understand the results. There are few typo and grammatical error that needs careful revision.
